# Dissecting phenotypic transitions in metastatic disease via photoconversion-based isolation

Yogev Sela[1,2,3†], Jinyang Li[1,2,3†], Paola Kuri[2,4], Allyson J Merrell[1,2,3], Ning Li[5,6], Chris Lengner[2,5,6,7], Pantelis Rompolas[2,4], Ben Z Stanger[1,2,3,6,7]*

[1]Department of Medicine, University of Pennsylvania, Philadelphia, PA, United States; [2]Department of Cell and Developmental Biology, University of Pennsylvania, Philadelphia, PA, United States; [3]Abramson Family Cancer Research Institute, University of Pennsylvania, Philadelphia, PA, United States; [4]Department of Dermatology, University of Pennsylvania, Philadelphia, PA, United States; [5]Department of Biomedical Sciences, School of Veterinary Medicine, Philadelphia, PA, United States; [6]Institute for Regenerative Medicine, University of Pennsylvania, Philadelphia, PA, United States; [7]Abramson Cancer Center, University of Pennsylvania, Philadelphia, PA, United States

*For correspondence:
bstanger@upenn.edu

[†]These authors contributed equally to this work

Competing interests: The authors declare that no competing interests exist.

**Abstract** Cancer patients often harbor occult metastases, a potential source of relapse that is targetable only through systemic therapy. Studies of this occult fraction have been limited by a lack of tools with which to isolate discrete cells on spatial grounds. We developed PIC-IT, a photoconversion-based isolation technique allowing efficient recovery of cell clusters of any size – including single-metastatic cells – which are largely inaccessible otherwise. In a murine pancreatic cancer model, transcriptional profiling of spontaneously arising microcolonies revealed phenotypic heterogeneity, functionally reduced propensity to proliferate and enrichment for an inflammatory-response phenotype associated with NF-κB/AP-1 signaling. Pharmacological inhibition of NF-κB depleted microcolonies but had no effect on macrometastases, suggesting microcolonies are particularly dependent on this pathway. PIC-IT thus enables systematic investigation of metastatic heterogeneity. Moreover, the technique can be applied to other biological systems in which isolation and characterization of spatially distinct cell populations is not currently feasible.

## Introduction

Metastasis is the primary cause of cancer-associated mortality and remains a significant therapeutic challenge. A considerable fraction of metastasis is occult (*Haeno et al., 2012*; *Vanharanta and Massagué, 2013*) and thus serves a potential source for residual disease and recurrence (*Sosa et al., 2014*; *Tohme et al., 2017*). While fluorescent reporters allow detection of metastatic colonies in pre-clinical models (*Aiello et al., 2016*; *Fluegen et al., 2017*), an inability to recover pure micron-scale colonies has precluded systematic phenotypic analysis of early-stage metastases. Several methods have been developed to isolate cells from specific compartments in vivo, laser-capture microdissection representing the most widely used platform (*Basnet et al., 2019*; *Espina et al., 2006*; *Lovatt et al., 2014*; *Tang et al., 2009*). However, the use of these methods is restricted to compartments of particular size and only for specific applications due to several limitations, including (1) contamination by undesired cells within the capture field; (2) low throughput; (3) a need for expensive and temperamental hardware; (4) loss of cell viability; and (5) lack of control over population composition. Spatial transcriptomics methods continue to improve and can distinguish local patterns across

large tumor regions (*Moncada et al., 2020*), but resolution limits and low representation of small metastatic colonies in the tissue hinders their isolation. Photoactivable and photoconvertible proteins provide a promising alternative for targeted cell isolation (*Medaglia et al., 2017*; *Nicenboim et al., 2015*). However, existing systems require specialized equipment and extended handling times, thus limiting scalability and precluding effective acquisition of rare or sporadic cells from multiple compartments such as microcolonies. Here, we report 'PIC-IT' ('Photomark and Isolate Cells If Tiny'), which enables unbiased and efficient isolation of size-defined metastatic colonies from live tissues through photoconversion-based marking (*Figure 1A*).

## Results

### Photoconversion-based isolation of size-specific metastatic colonies

Photoconvertible proteins comprise a class of fluorescent proteins whose emission switches from green to red upon exposure to blue light. For these studies, we used a version of the photoconvertible protein Dendra2 fused to H2B (H2B–Dendra2). As shown in *Figure 1—figure supplement 1A*, exposure of H2B–Dendra2-expressing pancreatic tumor cells (5074 cell line) to a mercury lamp-generated violet light (400–450 nm) for 30 s efficiently marks all cells within the field of view (FOV). Next, we engineered a highly metastatic pancreatic cancer cell line (6419 c5) to express H2B–Dendra2. Use of a clonal cell line was chosen to minimize genetic-based contribution to metastatic heterogeneity (*Hunter et al., 2018*). To generate metastasis, $10^4$ 6419c5–H2B–Dendra2 cells were injected into the pancreas of immune competent C57BL/6 mice, and tumors were allowed to grow for 3–5 weeks, at which point livers were examined under a standard widefield microscope.

Imaging revealed metastatic foci of various sizes, ranging from solitary tumor cells to micrometastases and larger macrometastases (*Figure 1B*, left). Importantly, strong nuclear fluorescence in all tumor cells enabled easy detection of metastatic foci of a particular size even with a low magnification 4× objective, allowing for rapid identification of regions of interest. To test whether region-specific photoconversion could be achieved using a widefield microscope in unsectioned, intact tissue specimens, we confined the FOV using the field-stop element (*Figure 1B*, 'Preconversion') and exposed cells to 400–450 nm light (*Figure 1B*, 'Photoconversion'). Exposure for <5 s (40× objective, N.A. = 0.75) led to a robust induction of red fluorescence in all cancer cells within the boundaries of the limited FOV (*Figure 1B*, 'Postconversion'). In a different FOV, using a 10× objective, we rapidly photoconverted (15 s, N.A. = 0.4) hundreds of cancer cells within a macrometastasis (*Figure 1—figure supplement 1B*). Spatial precision of labeling was achieved even when a low magnification objective was used, with a rapid drop-off of photoconversion outside the FOV boundary (*Figure 1—figure supplement 1C*). Importantly, background red fluorescence in non-converted Dendra2–H2B$^+$ cells was low, indicating minimal spontaneous conversion in our experimental model (*Figure 1B,D*, top left). These results demonstrate that both small and large metastatic lesions can be efficiently photoconverted using a widefield microscope by adjusting the FOV, exposure time, and numerical aperture.

We next tested whether we could recover photoconversion-marked metastatic cells for flow cytometry-based applications. To maximize accessibility to metastatic foci, we dissected the liver into millimeter-sized fragments and photoconverted cells within each fragment separately (*Figure 1C*). Subsequently, each liver fragment was designated for photoconversion of microcolonies (μCol, *Figure 1C*, top) or macrometastases (*Figure 1C*, middle). In parallel, cells in the primary tumor were also photoconverted (*Figure 1C*, bottom). This strategy allowed samples to remain chilled throughout the procedure, with exposure to ambient temperatures only during the photoconversion process itself (<6 min). Microcolonies were defined as collections of 10 or fewer metastatic cells, with the majority being one or two cells (*Figure 1—figure supplement 2A*), while macrometastases were defined as lesions of >400 μm in diameter. Following flow sorting, photoconverted cells derived from the different groups showed strong, comparable increases (~30-fold) in red fluorescence intensity (*Figure 1D*, *Figure 1—figure supplement 2B*). Although red fluorescence increased with longer exposure times, 5 s was sufficient to permit detection of photoconverted cells (*Figure 1—figure supplement 2C*). These results indicate that tissue resident H2B–Dendra2-expressing tumor cells can be readily detected by flow cytometry following brief photoconversion and tissue dissociation.

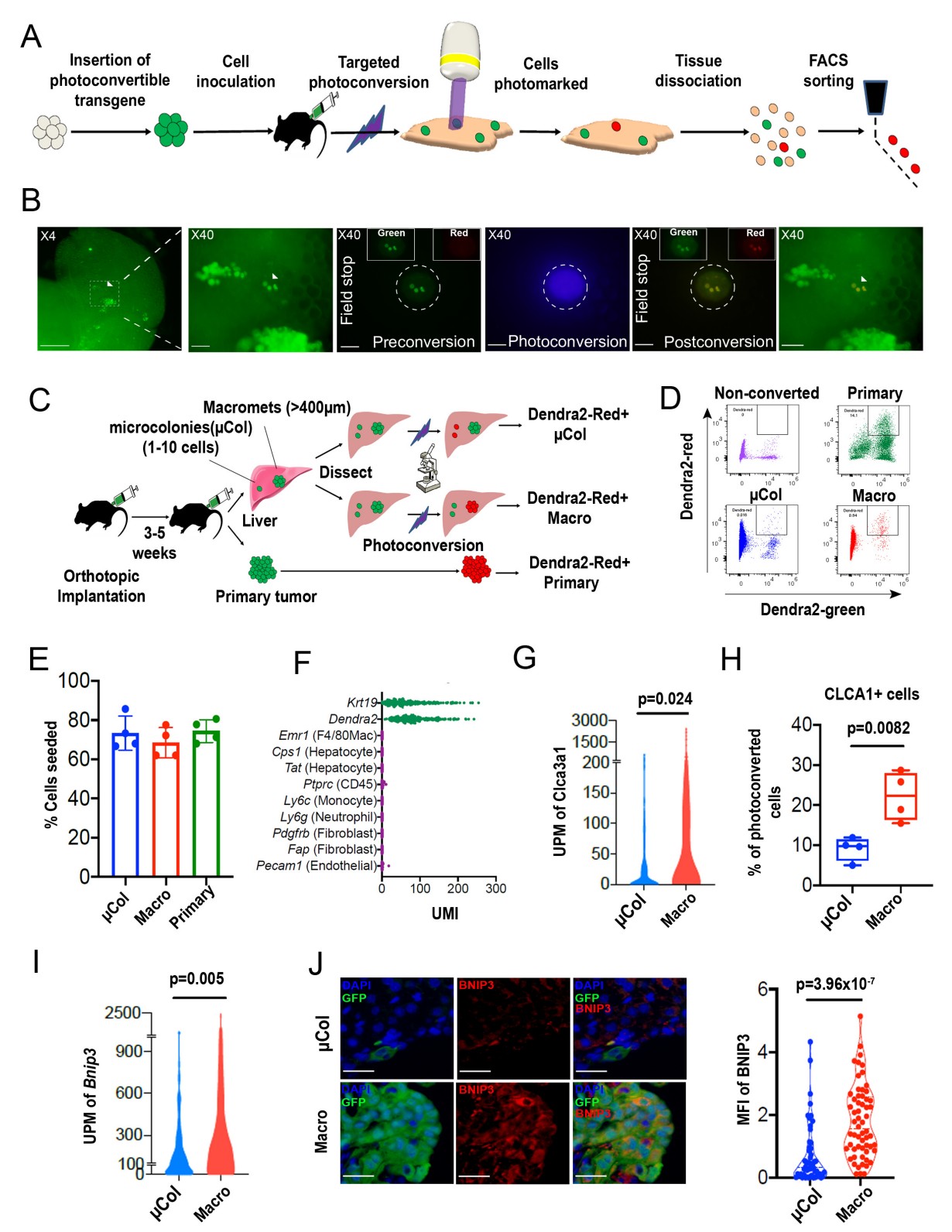

**Figure 1.** PIC-IT allows isolation of size-defined microcolonies and macrometastases from live animals. (**A**) Schematic of the PIC-IT protocol. (**B**) Photoconversion of Dendra2 in microcolonies spontaneously arising in the liver of a pancreatic cancer tumor model. Left to right: Microscopic views acquired through a 4× objective with region of interest containing three cell clusters destined for photoconversion located in the marked area (arrows denote the three cells of interest, scale bar = 250 μm). Focus on the marked area with 40× objective (scale bar = 25 μm). Field-stop-confined FOV

*Figure 1 continued on next page*

*Figure 1 continued*

within the 40× objective region, encircling the targeted three cell focus prior to photoconversion ('Preconversion'). In boxes, images representing individual green and red channels for the field-stop FOV. Field-stop image of a photoconversion session (violet light exposure, 'Photoconversion'). Field-stop confined FOV of an 40× objective following photoconversion ('Postconversion'). The 40× region with a full FOV demonstrating specificity of photoconversion to field-stop confined area. (C) Experimental scheme utilized for photoconversion-based isolation of metastatic cells from liver fragments. Orthotopic tumors derived from Dendra2-expressing cells produce liver metastasis. In different fragments, microcolonies (μCol) or macrometastases are photoconverted. Dissociated tissue from each group are pooled and sorted for subsequent applications. (D) Flow cytometry charts showing expression of Dendra2-green (X-axis) and Dendra2-red (Y-axis) in converted tumor cells and non-converted controls. (E) Seeding efficiency of photoconverted tumor cells, calculated as the fraction of cells counted in tissue culture plates 24 hr post-isolation relative to the number sorted by FACS (n = 4). (F) UMI profiles indicate expression of the markers *Krt19* and *Dendra2* in the sorted cells but no expression of other lineage markers (N = 361). (G, H) High prevalence of RNA-seq transcripts of *CLCA3A1* (G, $N_{\mu Col}$ = 94, $N_{macro}$ = 111) and the corresponding CLCA1 protein expression (H) determined by flow cytometry on macrometastasis-derived vs. microcolonies (n = 4). (I, J) Concordance between elevated transcription (UMI per million) of BNIP3 in RNA-seq (I) and (J) BNIP3 MFIs in (left) immunostained microcolonies and macrometastases populating the livers of 6419c5-YFP tumor-bearing mice ($N_{\mu Col}$ = 53, $N_{macro}$ = 60). On the right, representative images of microcolonies and macrometastases immunostained for BNIP3 (red), YFP (green), and DAPI (blue). Scale bar = 50 μm. Bars represent mean ± SEM in all graphs. p-values were calculated by unpaired two-tailed Student's t-test.

The online version of this article includes the following figure supplement(s) for figure 1:

**Figure supplement 1.** Spatial flexibility of PIC-IT.
**Figure supplement 2.** Kinetics of cell marking and efficiency of isolation using PIC-IT.
**Figure supplement 3.** Viable isolation of tumor cells by PIC-IT.

Because low throughput has been one of the key limitations impeding studies of microcolonies, we next sought to quantify the throughput of our method. To this end, we photoconverted 200–400 microcolonies in each of 16 different isolation sessions (~2.5 hr per session) and determined the yield of photoconverted tumor cells by flow cytometry. Across these sessions, recovery of photoconverted microcolonies (Dendra-red+ cells detected by flow cytometry as a % of total photoconverted cells quantified by microscopy) averaged at 38% (*Figure 1—figure supplement 2D*). Thus, a moderate output of 80–170 cells can be acquired per hour using this experimental setup (*Figure 1—figure supplement 2E*). Notably, we typically used less than 30% of the liver in our marking sessions; consequently, cell yield could potentially be improved with longer sessions, as there was minimal decay in the red signal up to 7 hr post-photoconversion (*Figure 1—figure supplement 2F*). Cells remained viable after flow sorting, with a ~70% seeding efficiency measured 24 hr after plating (*Figure 1E*). Importantly, cells sorted following photoconversion showed no evidence of DNA damage (*Figure 1—figure supplement 3A*) or impairment of colony formation (*Figure 1—figure supplement 3B*). Overall, these results demonstrate robust acquisition of hundreds of live cells from size-defined metastatic foci within a practical timeframe.

Single-cell RNA sequencing of 361 metastatic and primary tumor cells via Cel-seq2 (*Hashimshony et al., 2016*) revealed expression of dendra2 and *Krt19* but not markers of hepatocytes, endothelial cells, immune cells, and fibroblasts (*Figure 1F*), confirming the purity of the isolated cell populations. To further validate this RNA-seq expression analysis, we identified two genes – *Clca3a1* and *Bnip3* – whose expression was predicted to be significantly higher in macrometastases than microcolonies. We then confirmed this difference in expression by flow cytometry and immunostaining (*Figure 1G–J*). Taken together, our findings suggest that PIC-IT enables efficient, pure isolation of size-defined metastatic colonies, thereby enabling discovery of genes whose expression differs in the two populations.

## PIC-IT identifies enrichment for a pro-inflammatory phenotype in liver microcolonies

In various experimental settings, colonizing metastatic cells have been associated with physiological adaptations related to cellular state preceding dissemination (*Fluegen et al., 2017*) and/or the surrounding microenvironment (*Linde et al., 2016*; *Milette et al., 2017*). However, a systematic analysis of the phenotypic diversity of spontaneously arising microcolonies is currently lacking. We thus examined the transcriptional profiles of PIC-IT-extracted microcolonies and macrometases by Seurat clustering analysis (*Stuart et al., 2019*) and visualized their features by UMAP dimensional reduction (*McInnes et al., 2018*). At a global level, the distribution of microcolony- and macrometastatic-

transcriptomes broadly overlapped (*Figure 2A*). UMAP further identified three clusters in microcolonies and macrometastases, annotated as proliferative, inflammatory, or hypoxic/stressed, suggesting that microcolonies are phenotypically diverse (*Figure 2B*, *Figure 2—figure supplement 1A*). Notably, the inflammatory cluster, defined by elevated IFN-γ response and complement activity

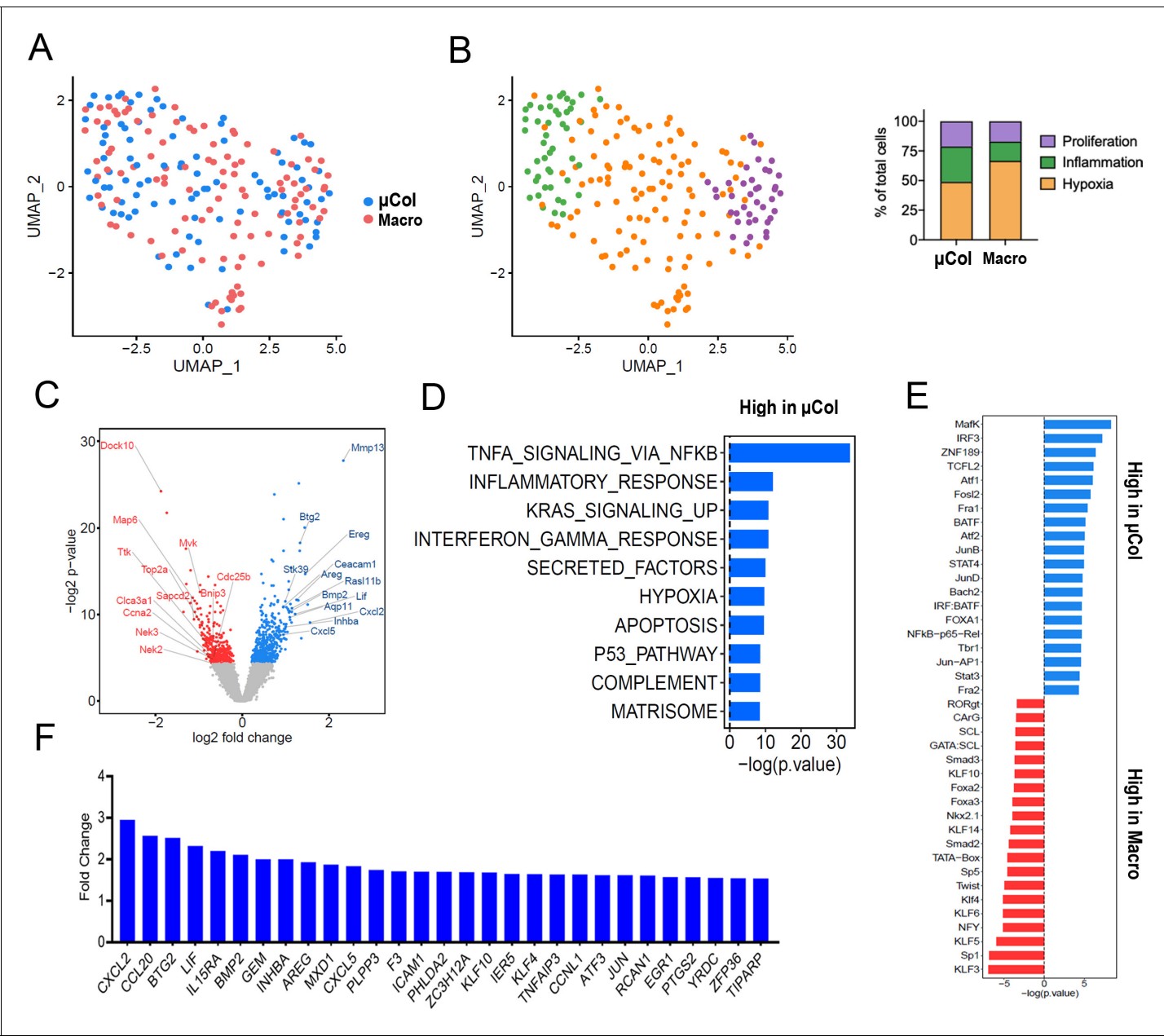

**Figure 2.** Phenotypic transitions in pancreatic tumor cells colonizing mouse liver. (**A**) Transcriptome distribution visualized by UMAP. (**B**) Distribution of metastatic cells for the different clusters functionally annotated by GSEA msGDMIB (p<0.05, 200 highest fold-change genes). On the right, proportions of clusters in microcolonies and macrometastases. (**C**) Volcano plot of differentially expressed genes in microcolonies and large metastasis derived from differential gene analysis (EdgeR). (**D**) Functional annotation of genes upregulated in macrometastasis. GSEA msGDMIB for Hallmark and canonical pathways (p<0.05, 200 highest fold-change genes). (**E**) Motif analysis of potential transcriptional regulator using HOMER with −1000 bp to +300 bp as scanning region and p<0.05, 200 highest fold-change genes as input (showing known results with p<0.01). (**F**) Fold-change increase (microcolonies vs. macrometastases) in differentially expressed genes (p<0.05) associated with Hallmark TNF-α-signaling-via-NF-κB signaling category derived from GSEA msGDMIB analysis.

The online version of this article includes the following figure supplement(s) for figure 2:

**Figure supplement 1.** Phenotypic transition from microcolony to macrometastasis.

(*Figure 2B*, *Figure 2—figure supplement 1B*), was enriched in microcolonies compared to macro-metastases (30% vs. 16%). In contrast, both the inflammatory and hypoxic clusters displayed decreased abundance of proliferation-related transcripts including CCNA2, TOP2A, CDK1, Ki-67, and PCNA (*Figure 2—figure supplement 1C*), consistent with previous observations on growth arrest mediated by hypoxia in colonizing metastatic cells (*Fluegen et al., 2017*). Next, we conducted a differential gene analysis using EdgeR (*Robinson et al., 2010*), comparing the transcriptomes of 94 microcolonies to 111 macrometastatic cells. This revealed 500 genes that were more highly expressed in microcolonies and 366 genes that were more highly expressed in macrometastases (p<0.05, *Figure 2C*). Functional annotation of the microcolony-associated genes revealed a strong signal for TNF-α-related NF-κB signaling and a prominent increase in inflammatory cytokine secretion (*Figure 2D*). Likewise, an IFN-γ-response signature was enriched in microcolonies, consistent with a pattern of enriched inflammatory gene expression. Furthermore, transcription factor motif analysis (HOMER) revealed enrichment of pro-inflammatory DNA regulatory elements in microcolonies (*Figure 2E*). In addition to the known inflammation-related transcription factors NF-κB, JUN/AP-1, and STAT, motif analysis also revealed enrichment of binding sites for MAFK, which has recently been demonstrated to enhance the transcriptional response to NF-κB (*Hwang et al., 2013*). In line with this enrichment of TNF-related NF-κB signaling, 29 genes associated with the Hallmark 'TNF-α-signaling-via-NF-κB' gene set were expressed at significantly higher levels in microcolonies compared to macrometastatic cells (*Figure 2F*, *Figure 2—figure supplement 1D*). Taken together, transcriptional profiling suggests that pancreatic microcolonies in the liver manifest a strong pro-inflammatory phenotype associated with increase in NF-κB and AP-1 signaling.

## Microcolonies exhibit reduced propensity to proliferate

We next turned our attention to the genes with low expression in microcolonies relative to macrometastases. Intriguingly, functional annotation indicated enrichment for cell cycle and cholesterol biosynthesis pathways in the macrometastases (*Figure 3A*), with elevated expression of a number of mitosis regulators such as *Nek2* (*Fry et al., 2012*), *Sapcd2* (*Xu et al., 2007*), and *Ttk* (*Dominguez-Brauer et al., 2015*, *Figure 3B*). Consistent with this, microcolonies exhibited higher expression of negative regulators of cell cycle progression including *Ceacam1* (*Sappino et al., 2012*), *Stard13* (*Jaafar et al., 2020*), and *Mir22hg* (*Xu et al., 2020*, *Figure 3B*), predicting an overall hypo-proliferative state for microcolonies.

To determine the proliferative state of the cells, we profiled microcolonies and macrometastases for expression of cell cycle markers by immunofluorescence. Ki-67 immunostaining revealed no measurable difference in the percentage of positive cells in microcolonies vs. macrometastases (*Figure 3C*), consistent with our previously reported findings (*Aiello et al., 2016*). However, the frequency of phosphorylated-histone H3-positive cells was lower in microcolonies (*Figure 3D*), suggesting fewer mitotic events. Examination of cell cycle suppressors indicated comparable frequencies of pRB-positive cells across metastatic stages (*Figure 3E,F*). However, expression levels of pRB were consistently higher in macrometastatic cells, indicating higher propensity for cell cycle progression in these cells (*Figure 3G*). Additional staining revealed a trend toward higher prevalence of p21 in microcolonies (*Figure 3H*). Taken together, these findings suggest that the cycling fractions are similar in microcolonies in macrometastatic cells, but rate of transit through the cell cycle may be slower in microcolonies.

We hypothesized that cells in larger metastatic lesions may be inherently 'poised' to proliferate more readily than those in smaller metastatic lesions, a difference that can be challenging to test in vivo. We therefore exploited PIC-IT's capacity to isolate live cells and functionally measured the proliferative potential of the different metastatic groups. Specifically, microcolony- and macrometastasis-derived cells were extracted from mice bearing 6419c5-H2B-Dendra2 tumors, seeded at a density of 45 cells/well, and monitored daily for colony growth (*Figure 3I*). While cells derived from microcolonies and macrometastases expanded to a similar extent after 24 hr, colonies derived from macrometastases were larger at 48 hr and 72 hr (*Figure 3J,K*). These results support our molecular profiling results and suggest that macrometastatic cells are primed for cell division (or, conversely, that microcolonies are primed for cell cycle arrest).

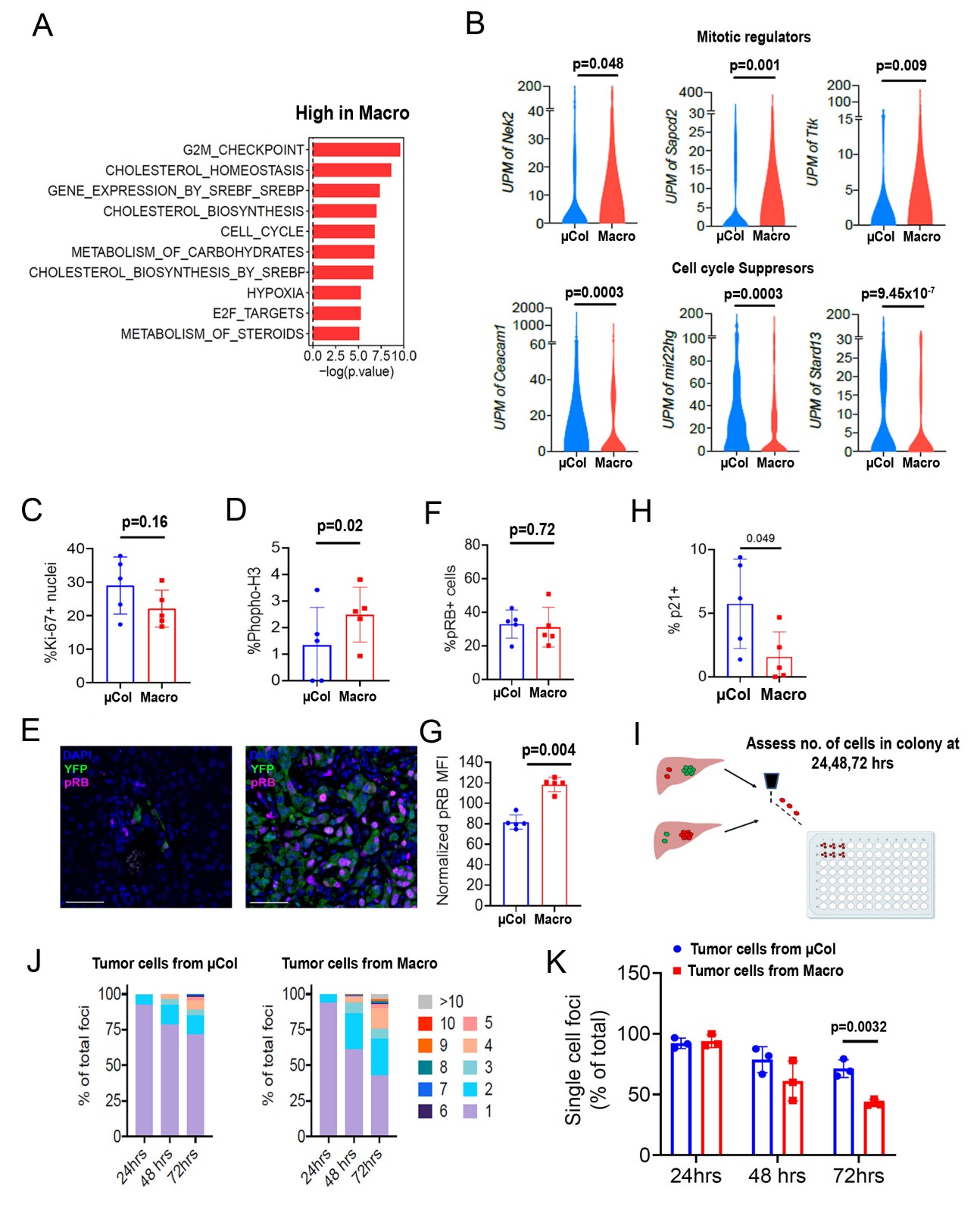

**Figure 3.** Microcolonies exhibit a hypo-proliferative phenotype. (A) Functional annotation of genes downregulated in microcolonies/higher in macrometastases. GSEA msGDMIB for Hallmark and canonical pathways (p<0.05, 200 highest fold-change genes). (B) Abundance of transcripts in RNA-seq for genes associated with cell cycle arrest (Top) and positive regulation of mitosis (Bottom). ($N_{\mu Col}$ = 94, $N_{macro}$ = 111). Fraction of Ki-67-positive cells (C) and phosphorylated-histone H3-positive cells (D) in microcolonies and macrometastases as determined by immunostaining (n = 5 samples per

*Figure 3 continued on next page*

*Figure 3 continued*

group). (**E**) Representative confocal images of microcolonies and macrometastases from livers of 6419 c5 mice stained with anti-phospho-Rb (pRB; Ser807/811). Scale bar = 50 µm. (**F**) Fraction of pRB+ microcolonies and macrometastases, and (**G**) RB phosphorylation levels in pRB-positive cells normalized to average MFI in each mouse (n = 5 samples per group). (**H**) Fraction of p21+ cells in microcolonies and macrometastases as determined by immunostaining (n = 5 samples per group). (**I**) Schematic for ex vivo proliferation assay. Microcolonies and macrometastases are extracted via PIC-IT, seeded at 45 cells/96 wells and colony growth is monitored via microscopy over 72 hr. (**J**) Colony size distribution of metastatic cells over the first 72 hr after isolation. Left panel, colonies derived from microcolonies. Right panel, colonies derived from macrometastases (n = 3 independent experiments). (**K**) Fraction of isolated metastatic cells residing as solitary cells at 24, 48 and 72 hr post-isolation (n = 3 independent experiments). Bars represent mean ± SEM in all graphs. p-values were calculated by unpaired two-tailed Student's t-test.

## Microcolonies require NF-κB /AP-1 signaling

We previously reported that solitary pancreatic tumor cells exhibit resistance to standard-of-care chemotherapy (gemcitabine + nab-paclitaxel) in autochthonous KPCY mice (*Aiello et al., 2016*). Consequently, we sought to identify vulnerabilities that might be specific to tumor cells in the earliest stages of colonization. As our profiling experiments indicated enrichment of a pro-survival inflammatory response in microcolonies, we hypothesized that this pathway – driven by NF-κB and/or AP-1 – might be important for the persistence of these cells in vivo. First, we performed NF-κB-p65 staining in the 6419 c5 implanted cell line model and the KPCY autochthonous model. These two models confirmed that microcolonies exhibit higher levels of nuclear NF-κB-p65 compared to macrometastases (*Figure 4A,B*). In addition, and as suggested from our motif analysis, phospho-c-Jun-positive cells were more prevalent in microcolonies than in macrometastases in both models (*Figure 4— figure supplement 1A,B*). Expression of NF-κB-p65, and to a lesser degree phospho-c-Jun, was also enhanced in microcolonies of KPCY mice lacking overt metastatic disease (*Figure 4—figure supplement 1B,C*), suggesting that the pro-inflammatory phenotype does not depend on the presence of large metastases. These findings suggest that the enhanced inflammatory signatures identified through PIC-IT-enabled RNA sequencing reflect true signaling differences in small and large metastases.

To functionally test whether this pro-inflammatory response is required for microcolony viability, we tested the activity of triptolide (TPL), a potent anti-inflammatory agent that inhibits both pathways (*Yuan et al., 2019*). We inoculated 6419c5-YFP cells into the pancreas and allowed implants to grow for 21 days (*Figure 4C*), a time point at which >95% of mice develop macrometastases (data not shown). Animals were then randomized to receive either triptolide (0.2 mg/kg) or vehicle by intraperitoneal injection (7 d) followed by quantification of microcolonies and macrometastases by immunohistology (*Figure 4C*). Although the total liver area occupied by metastases (dominated by macrometastases) was similar in the two treatment groups (*Figure 4D*), triptolide treatment led to a significant reduction in the relative frequency of microcolonies (*Figure 4E*) and the ratio of microcolonies or single-metastatic cells to macrometastases (*Figure 4F,G*). These results are in line with a recent report highlighting the potential of neoadjuvant treatment with NF-kB-targeting therapy in PDAC (*Saito et al., 2019*) and suggest that microcolonies are more susceptible to inhibition of inflammatory signaling than macrometastases.

## Discussion

Studying the occult fraction of metastatic disease is challenging due to the rarity of cells comprising micrometastatic lesions and a lack of efficient tools to isolate and study them. PIC-IT overcomes these challenges and permits the isolation and systematic characterization of cells comprising both large and small metastatic lesions. Our experiments reveal that microcolonies are a diverse cell population enriched with cells manifesting a pro-inflammatory response. Our findings further highlight a specific role for NF-κB/AP-1-regulated inflammatory pathways in microcolonies not shared by the majority of metastatic cells (macrometastases), demonstrating the utility of our approach in identifying susceptibilities of a micro-fraction of the metastatic population that drives disease recurrence and mortality. Inflammation plays a key role in tumor initiation and progression in PDAC and has also been implicated in metastasis formation (*Stone and Beatty, 2019*). Non-uniform NF-kB signaling across the metastatic spectrum in our study is somewhat surprising and could potentially reflect altered levels of TNF-α in the microenvironment of a growing metastasis (*Milette et al., 2017*) or a

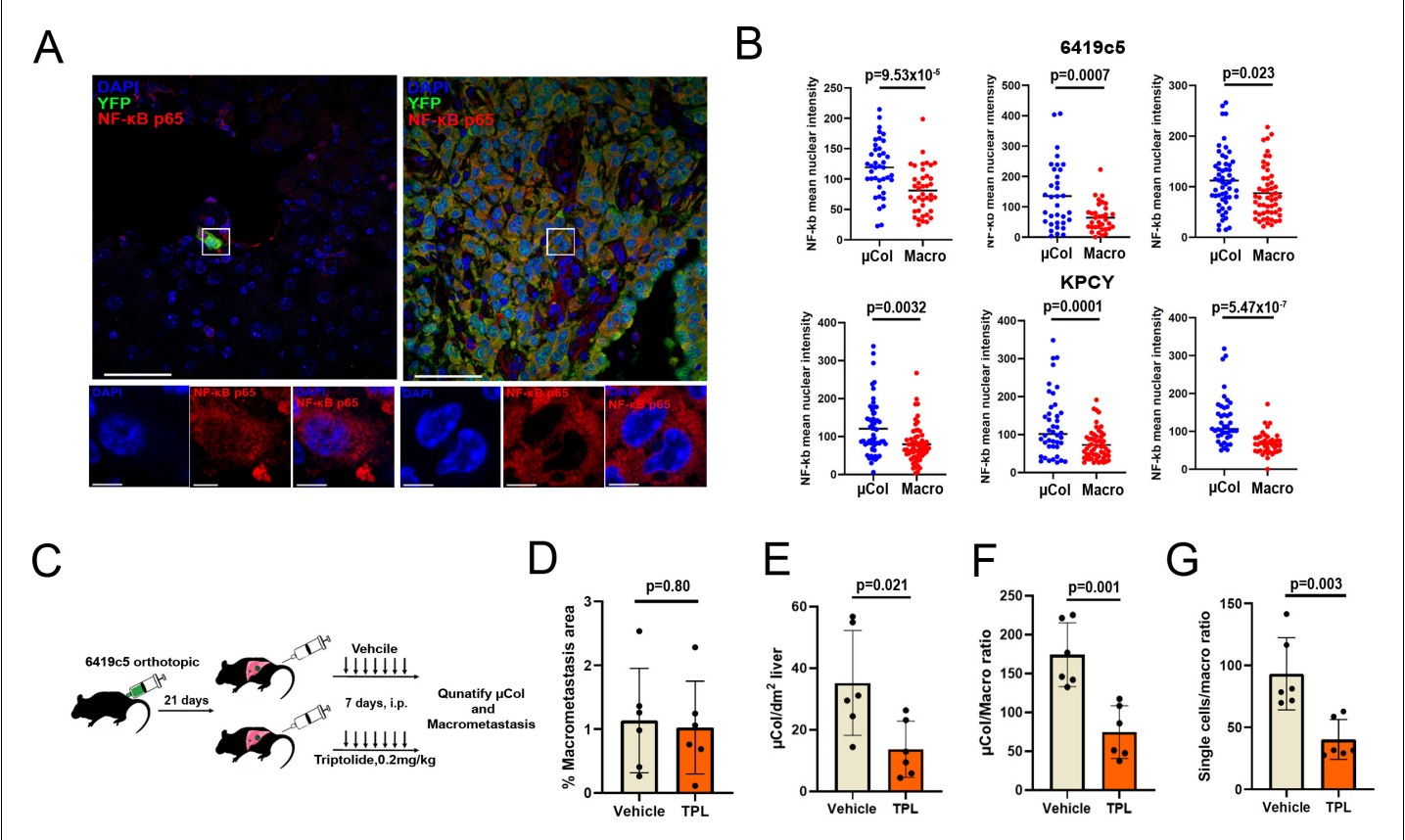

**Figure 4.** Microcolonies are susceptible to inhibition of NF-κB /AP-1 signaling. (**A**) Representative confocal images of microcolonies and macrometastases from livers of KPCY mice stained with NF-κB (red) and YFP (green) antibodies. Scale Bar = 50 μm. On the bottom, higher magnification inset showing nuclear signal of NF-κB in 1 μm z-section, with NF-κB (red) and DAPI (blue). Scale bar = 5 μm. (**B**) Quantification of mean nuclear intensity of NF-κB in microcolonies and macrometastases of orthotopic 6419c5-YFP (top) and KPCY mice (bottom). Each dot represents one animal. (**C**) Experimental scheme for targeting NF-κB /AP-1 in established tumors with triptolide (TPL). (**D–G**) Quantification of metastatic colonies in livers of TPL- and vehicle-treated mice. (**D**) Percentage area of liver occupied by macrometastases. (**E**) Absolute number of microcolonies per unit area in liver sections. (**F**) Ratio of microcolony frequency to macrometastatic area. (**G**) Ratio of metastatic single-cell frequency to macrometastatic area. (For D–G, n = 6 animals per group, 10 liver sections sampled for each animal.) Bars represent mean ± SEM in all graphs. p-values were calculated by unpaired two-tailed Student's t-test.

The online version of this article includes the following figure supplement(s) for figure 4:

**Figure supplement 1.** Increased activity of c-JUN and NF-kB in microcolonies.

pre-existing 'NF-kB' activity associated with tumor dissemination. Of note, this latter possibility is hinted at by marked upregulation of TNF-α-NF-kB signaling in PDAC CTCs (*Ting et al., 2014*). Nonetheless, as isolated microcolonies may originate in either primary tumor or metastasis, it remains to be determined whether the pro-inflammatory profile reflects the pre-metastatic state of the 'seed' or an acquired adaptation to the 'soil'.

Our study also indicated differences in proliferation propensity between tumor cells of microcolonies and macrometastases. Transcriptional profiles and immunohistochemistry profiling together with direct measurement of proliferation capacity ex vivo enabled by PIC-IT, suggested that macrometastases are more 'poised' for proliferation compared to microcolonies. Comparable rates of Ki-67 and pRB staining suggest that microcolonies and macrometastatic cells do not differ in their commitment for cell cycle entry. Rather, as implied by their characteristically lower pRB phosphorylation levels and phospho-histone H3+ fractions, cells in microcolonies are less committed to cell cycle progression and mitosis. Although the mechanism underlying reduced pRB phosphorylation and hypo-proliferative state in microcolonies is unclear, a recent study demonstrated that TNF-α-

mediated NF-kB signaling can specifically attenuate cell cycle transition through the G1/S boundary (*Ankers et al., 2016*).

While the current study used PIC-IT to isolate cells at different stages of metastasis, the method has many potential applications. For example, it could be used to isolate cells in different 'niches' of the tumor microenvironment (e.g. cells in well perfused vs. poorly perfused regions), thus paving the way for testing functional features such as proliferation, migration, metabolic characteristics, and drug resistance in heterogeneous tumor cell populations. As our identification of CLCA1 shows, the technique can be used to identify protein markers – particularly those residing on the cell surface – with which to prospectively isolate cells of interest by flow cytometry. Finally, the technique is particularly well suited to the isolation of spatially defined rare cells – such as stem cells – in malignant and benign conditions.

## Materials and methods

### Cell culture

Pancreatic tumor cells were isolated from late-stage primary tumors from C57BL/6 background KPCY mice and generated by limiting dilution as described (*Li et al., 2018*). All mouse pancreatic tumor cell clones were tested by the Research Animal Diagnostic Laboratory (RADIL) at the University of Missouri, using the Infectious Microbe PCR Amplification Test (IMPACT) II. Tumor cells were cultured in Dulbecco's modified Eagle's medium (DMEM) (high glucose without sodium pyruvate) with 10% fetal bovine serum (FBS) (Gibco) and glutamine (2 mM). The clones were regularly tested using the MycoAlert Mycoplasma Detection Kit (Lonza).

### Genetic modification of tumor cells

6419c5-YFP cells were transduced with a sgRNA targeting endogenous YFP to remove its expression (sgRNA sequence: GGGCGAGGAGCTGTTCACCG). Subsequentially, 6419 c5 and 5074 cells were transduced with a lentiviral expression vector of h2b-dendra2 (Addgene #51005). Single-cell clones expressing Dendra2-H2B were derived by FACS sorting.

### Animals

Wild-type C57BL/6 were purchased from The Jackson Laboratory and/or bred at the University of Pennsylvania. For tumor-bearing mice, end point criteria included severe cachexia, weakness, and inactivity.

### Orthotopic implantation of tumor cells

Tumor cells were dissociated into single cells with 0.25% trypsin (Gibco), washed with serum-free DMEM twice, and counted in preparation for orthotopic implantations. Three thousand to 10,000 tumor cells were implanted orthotopically into 6–8 week old female or male C57BL/6 mice. Tumors were harvested 3–5 weeks following implantation.

### Triptolide treatment

Ten thousand 6419c5-YFP tumor cells were implanted orthotopically, and tumors were allowed to grow for 3 weeks. Mice were randomized into two groups and injected intraperitoneally with either 0.2 mg/kg triptolide (HY-32735, Medchem Express) or vehicle (10% DMSO, 40% PEG300, 5% Tween-80, 45% saline) for seven consecutive days. Livers were sectioned, and microcolony frequency represents 10 sections per mouse. Macrometastatic area and total liver area were measured using imageJ.

### PIC-IT

Tumor-bearing mice were sacrificed, and livers or tumors were removed and fragmented using scissors to small pieces (~2–5 mm in diameter) and kept in flow buffer (HBSS) on ice. Fragments were further flattened and mounted on top of a Petri dish, covered with a drop of flow buffer (HBSS, 2.5 mM HEPES 5% FBS, 2 mM glutamine, 1% penicillin/streptomycin, 1% non-essential amino acids, 0.5%fungizone, 0.3% glucose, 20 mM MgCl$_2$, 1 mM sodium pyruvate) and examined under a BX60 Olympus upright microscope using a ×10 objective unless otherwise specified. Metastasis were

identified, and FOV was confined around metastatic foci of choice using the field-stop element. Cells were then photoconverted by illumination with a wavelength of 400–450 nm using mercury lamp and a 451/106 nm BrightLine Full Spectrum Blocking single-band bandpass filter (FF01-451/106-25, Avro Inc), and red fluorescence was confirmed. Photoconverted specimen of each metastatic group was pooled and transferred to a C-column (130-093-237, Miltentyi Biotech) and digested by incubation with 2 mg/ml collagenase IV+DNase I (Sigma) in DMEM using the mLIDK program of a gentleMACS (Miltentyi Biotech). Cells were mashed on 100 µm MESH strainer, pelleted, and dissociated to single cells using Accutase (07922, Stemcell Technologies) for 5′ and countered, pelleted, and incubated for 3′ with ACK lysis buffer (10-548E, Lonza) to remove red blood cells. Pelleted cells were resuspended with flow buffer containing 1 µg/ml DAPI + DNase I, incubated for 5′ in 37°C, filtered through 100 µm mesh strainer, and either analyzed by LSR-II or sorted using a FACSaria. Isolation experiments represent independent experiments performed on different days.

## Spatial analysis

For spatial analysis, primary tumors were processed through a VT1200S vibratome (Leica Biosystems) to 200 µm slices. One maximally confined FOV of ×10 objective was photoconverted per slice, and Z-stacks were acquired by a LEICA TCS SP8 microscope. Sum intensity projections from each sample were analyzed by ImageJ using the radial profile plugin.

## Flow cytometry and cell sorting

For flow cytometric analyses, dissociated livers were stained with Armenian hamster anti-CLCA1 1:5 (10.1.1, Iowa Hybridoma bank) for 25′ following ACK lysis. Samples were then washed and stained with an APC-conjugated secondary antibody and PE-Cy7-CD45 antibody (103114, Biolegend) and then analyzed by flow cytometry using BD FACS LSR-II (BD Biosciences) and analyzed FlowJo software (Treestar).

For cell sorting, digested organs were sorted into 96 wells prepped for cel-seq2 or 96-well culture dishes for re-culturing experiments using FACSariaII equipped with a green laser through a 100 µm nozzle. Isolation experiments represent three experiments performed on different days.

## Cel-seq2 profiling

Library preparation was performed by adhering to the developer's protocol using the original barcodes described (*Hashimshony et al., 2016* in *Genome Biology*). Libraries were quantified using qubit (ThermoFisher), and library fragmentation was assessed via tapestation (Agilent). Libraries were prepared for sequencing using Illumina 75 cycles high output kit v2.0 (20024096, Illumna) and sequenced on a NextSeq 500/550 with 26 bases on read 1 (R1), 8 bases for the Illumna index, and 41 bases for read 2 (R2).

## Immunofluorescent and immunohistochemistry staining

For immunostaining, tissues were fixed in Zn-formalin for 24 hr and embedded in paraffin. Sections were deparaffinized, rehydrated, and prepared by antigen retrieval for 6 min each, and then blocked in 5% donkey serum for 1 hr at room temperature, incubated with either Rabbit anti-BNIP3 antibody at 1:100 (A5683, Abclonal), Rabbit anti-NF-kB-p65 1:400 (8242S, Cell Signaling Technology), Rabbit anti-Phospho-c-jun 1:200 (3270S, Cell Signaling Technology), Rat anti-ki67 1:100 (14-5698-82, ebiosciences), Rabbit anti-pRB-Ser807/811 1:400 (8516, Cell Signaling Technology), anti-p21 1:1000 (ZRB1141, Sigma), Rat anti-Phospho H3 1:1000 (H9908, Sigma), or Goat anti-GFP (ab6673, Abcam) at 1:250 overnight at 4°C, washed, incubated with secondary antibody AF594-anti-rabbit (A-21207, Invitrogen), Alexa594 anti-rat (A-21209, Invitrogen), and Alexa488 anti-goat (11055, Invitrogen) antibodies at 1:200 for 1 hr at room temperature, counterstained with DAPI, washed, and mounted. Slides were visualized and imaged using an Olympus IX71 inverted multicolor fluorescent microscope and a DP71 camera using ×400 magnification. MFI was calculated by measuring average intensity over a fixed threshold for all images. NF-kB staining was imaged via Zeiss LSM880 confocal microscope using ×63 oil-immersion objective, and intensity was measured on 0.8 µm z-sections.

## Computational analysis

Fastq files were generated from bcl files using bcl2fastq tool from illumina. Reads were then trimmed to 12 bp for forward sequencing (R1) and 36 bp for reverse sequencing (R2), using cutadapt software (https://cutadapt.readthedocs.io/en/stable/). Trimmed reads were first debarcoded using UMI_tools with 'extract' function and default settings (https://github.com/CGATOxford/UMI-tools) and then aligned to mm10 genome using STAR (https://github.com/alexdobin/STAR). Aligned reads were counted using featureCounts (http://subread.sourceforge.net) and output as bam files. Generated bam files were then sorted and indexed using samtools. Molecules were counted using UMI_tools with 'count' function and default settings to generate UMI tables. The merged UMI table was subsequently imported into R-studio (R version 3.3.3) and used as input for edgeR analysis to assess differentially expressed genes. Samples with more than 5,000 UMI were used for differential gene expression analysis. Differentially expressed genes were also used as input file for online GSEA following provided instruction. Detailed Scripts and parameters used for each steps of analysis could be provided by reasonable request to the authors. Gene signatures were extracted by performing edgeR analysis.

For visualization of the single cell RNA-seq results, we used the Seurat package. We performed SCT transformation for data normalization and integration. We then performed linear dimensional reduction using PCA, followed by clustering using K-nearest-neighbor graph-based clustering and Louvain modularity optimization (with dimensions = 1:15 and resolution = 1), and non-linear dimensional reduction using UMAP method. Markers for each identified clusters were generated using Seurat with default settings and used as input for function pathway analysis, including online GSEA using the molecular signatures database.

For motif analysis, top differentially expressed genes were used as input for HOMER ($-1000$ bp to $+300$ bp as scanning region) for identification of motifs of potential transcriptional regulators.

## Ex vivo culture assay and colony formation

Sorted tumor cells were seeded at a density of 45 cells per 96 wells on growth factor-reduced Matrigel (BD) and allowed to grow for up to 6 days under standard culture conditions (see Cell culture). Retention was estimated 24 hr post-seeding. For colony size quantifications, cultures were evaluated daily, and the number of cells was manually counted per colony. Isolation experiments represent three experiments performed on different days.

## Software and statistical analysis

PRISM software and R were used for data processing, statistical analysis, and result visualization (http://www.graphpad.com). The R language and environment for graphics (https://www.r-project.org) was used in this study for the bioinformatics analysis RNA-seq. The R packages used for all analyses described in this manuscript were from the Bioconductor and CRAN. Statistical comparisons between two groups were performed using Student's unpaired t-test. For comparisons between multiple groups, one-way ANOVA with Tukey's HSD post-test was used. On graphs, bars represent either range or standard error of mean (SEM), as indicated in legends. For all figures, $p < 0.05$ was considered statistically significant.

## Acknowledgements

We acknowledge Priyanka Verma and Jelena Petrovic for technical assistance with IHC and next-generation sequencing. We acknowledge Cynthia Clendenin and Rina Sor from the pancreatic cancer mouse hospital for their assistance with animal studies. We thank Andrea Stout from the Molecular and the CDB Microscopy Core for technical assistance. We acknowledge Taiji Yamazoe, Robert Norgard, and Jason Pitarresi for technical support.

## Additional information

### Funding

| Funder | Grant reference number | Author |
|---|---|---|
| National Cancer Institute | CA229803 | Ben Z Stanger |
| National Institutes of Health | DK083355 | Ben Z Stanger |

The funders had no role in study design, data collection and interpretation, or the decision to submit the work for publication.

### Author contributions

Yogev Sela, Conceptualization, Data curation, Formal analysis, Validation, Investigation, Visualization, Writing - original draft, Project administration, Writing - review and editing; Jinyang Li, Data curation, Software, Formal analysis, Validation, Investigation, Visualization, Writing - review and editing; Paola Kuri, Formal analysis; Allyson J Merrell, Ning Li, Chris Lengner, Pantelis Rompolas, Writing - review and editing; Ben Z Stanger, Conceptualization, Resources, Supervision, Funding acquisition, Investigation, Writing - review and editing

### Author ORCIDs

Jinyang Li (iD) http://orcid.org/0000-0001-8125-6603
Ben Z Stanger (iD) https://orcid.org/0000-0003-0410-4037

### Ethics

Animal experimentation: This study was performed in strict accordance with the recommendations in the Guide for the Care and Use of Laboratory Animals of the National Institutes of Health. All of the animals were handled according to approved institutional animal care and use committee (IACUC) protocols (#804643) of the University of Pennsylvania.

### Decision letter and Author response

Decision letter https://doi.org/10.7554/eLife.63270.sa1
Author response https://doi.org/10.7554/eLife.63270.sa2

## Additional files

### Supplementary files

• Transparent reporting form

### Data availability

Sequencing data have been deposited in GEO under the accession code GSE158078.

The following dataset was generated:

| Author(s) | Year | Dataset title | Dataset URL | Database and Identifier |
|---|---|---|---|---|
| Sela Y, Li J, Stanger BZ | 2021 | Transcriptional profiling of metastatic tumor cells in liver of a mouse pancreatic cancer model | https://www.ncbi.nlm.nih.gov/geo/query/acc.cgi?acc=GSE158078 | NCBI Gene Expression Omnibus, GSE158078 |

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
