## [Decision Letter]

**Acceptance summary:**

In this short report, Sela, Li and collaborators describe an innovative methodology to track and isolate metastatic cells, be them single (which they call microcolonies) or within macrometastasis, after injection into an in vivo model. The method consists in the use of a photoconvertible protein which changes emission from green to red when put under blue light. The authors then use their method to describe the biological characteristics of microcolonies and macrometastasis in a murine pancreatic cancer model. This methodology is elegant and has high potential to be adopted by the scientific community due to its easiness and accessibility.

**Decision letter after peer review:**

Thank you for submitting your article "Dissecting phenotypic transitions in metastatic disease via photoconversion-based isolation" for consideration by *eLife*. Your article has been reviewed by three peer reviewers, including C Daniela Robles-Espinoza as the Reviewing Editor and Reviewer #1, and the evaluation has been overseen by Richard White as the Senior Editor. The following individual involved in review of your submission has agreed to reveal their identity: Patricia A Possik (Reviewer #2).

The reviewers have discussed the reviews with one another and the Reviewing Editor has drafted this decision to help you prepare a revised submission.

In this short report, Sela, Li and collaborators describe an innovative methodology that uses photoconversion of H2B-Dendra2, selective tissue processing and FACS to compare solitary disseminated tumor cells (DTCs) and macrometastasis (mainly) in a model of pancreatic cancer (PanC) metastasis to the liver. The reviewers agree that the manuscript is interesting and the work presented is generally sound. However, there are a number of issues that they have raised that would need to be addressed before it can be published. If the suggested experiments cannot be performed at this stage, then the conclusions and interpretation would need to be adjusted to address these comments.

Major points

(1) The authors isolate disseminated tumor cells (DTC) and metastasis from tissues where these cells co-exist. It is assumed but not proven that the solitary DTCs came there from the pancreas and remained solitary while others grew out to form the profiled metastasis. However, the profiled DTCs could have separated from the neighboring metastasis large or small. PanC cells can be quite motile as shown by the senior author before and can move long distances (Nature. 2018 Sep;561(7722):201-205.). Further, a caveat for the interpretation of the profiles is that the presence of large metastasis may already alter the micro-environment explaining why the Seurat analysis cannot separate solitary and met populations – it is highly unusual that DTCs and metastasis do not differ in some PCA analysis. Without an exclusive solitary DTC state analysis the true nature of the DTCs may not be accurately resolved.

To solve the above issue, the authors should perform two controls: (1) take advantage from the label retaining utility of Dendra2 or H2B-Dendra 2 (some papers are cited where this advantage was utilized) and photoconvert the cells in primary lesion. Then they could track and isolate red and green DTCs and later only green clusters and profile those using the same methods. (2) Option 1 is the best as it uses fate mapping reporting on quiescence and proliferation. But an alternative option is that the authors excise livers earlier and photoconvert solitary DTCs when there are still no metastasis. This would be very informative to distinguish the profiles of solitary DTCs in the context of low burden and more homeostatic liver vs. a high burden disease liver. If there are no differences between the DTCs then the isolation of stage IV-like DTCs becomes useful to profile those even in patients. But if there are differences then it may reveal how an earlier stage liver differs in the DTC populations from a late progression liver, information that would be very useful. Such analysis would enhance the paper as it remains limited in mechanism identification and strength of the data and phenotypes described.

(2) About the proliferative phenotype attributed to DTCs. Although single cell analysis suggests a non-proliferative phenotype for DTCs, no difference is observed in Ki67 staining. Then the authors show that isolated cells proliferate at different rates depending on their origin: DTCs were slower in increasing the number of cells in their colonies compared to cells isolated from metastatic colonies. On this single experiment, DTCs exhibit reduced propensity to proliferate, which for the authors was sufficient to suggest that these cells may be primed for cell cycle arrest. This conclusion needs more supporting information since no other marker of cell cycle arrest is observed. The authors see an increase cell cycle signatures in macrometastasis, but no evidence of cell cycle arrest, whether in gene signatures or cell culture experiments, is shown. What about phospho-histone 3 or P-Rb (by IF) or p27, p21, p16, p57 (IF or qPCR)? The re-culturing, while a nice assay, really shows no striking differences. There is a short lag difference that could be explained by adaptation or resolving DNA damage or alterations in the actin cytoskeleton as hinted from the ceacam and stard13 upregulation. The genes shown in Figure 3 are interesting but not classical negative regulators of growth.

(3) Application of DTC and macrometastasis scores to TCGA data and survival. This is an unusual comparison. The primary tumor is already a growing proliferative mass with a strong stromal component in a different microenvironment. The comparison reveals probably only proliferation in the pancreas or in overt metastasis, which appears to be true as the curves only separate in the first 20 months arguing that the authors are looking at effects on growth regardless of the DTC or macro-metastasis state. Perhaps metastasis free proportion over time should be tested but more importantly the signatures from the requested experiments should be tested. Is there RNAseq from metastasis available to explore these mechanisms? One would assume that the profiles of the DTCs may not appear there but the macro-metastasis would?

Regarding the survival analyses: Were these controlled for other patient characteristics? How do the authors explain that enrichment of DTC-associated genes predicts poor survival in patients as opposed to those genes enriched in macromets? One may think the alternative is more likely. This seems to agree with the finding that DTCs seem to have an hypoproliferative phenotype compared to macromets – so how do they influence survival?

(4) Figure 4: This experiment is negatively affected by the same issues as in Point 1. The DTCs could be derived from the metastasis and not really true originally arrived DTCs. This experiment would be more impactful showing that H2B-dendra photoconverted cells from the primary site are eliminated by Abs to the chemokines or knock-down of its receptors. This would be useful and one could imagine how patients with PCa may receive prophylactic treatments to reduce DTCs based on unique biology. In stage IV disease it is probably meaningless unless some drug causes massive reduction in overt metastatic disease.

---

## [Author Response]

In this short report, Sela, Li and collaborators describe an innovative methodology that uses photoconversion of H2B-Dendra2, selective tissue processing and FACS to compare solitary disseminated tumor cells (DTCs) and macrometastasis (mainly) in a model of pancreatic cancer (PanC) metastasis to the liver. The reviewers agree that the manuscript is interesting and the work presented is generally sound. However, there are a number of issues that they have raised that would need to be addressed before it can be published. If the suggested experiments cannot be performed at this stage, then the conclusions and interpretation would need to be adjusted to address these comments.

We are grateful for the overall positive response of the reviewers. We have edited the text and performed additional analyses to attend to the points they raise, resulting in <milestone-start />5<milestone-end /> new main figure panels and <milestone-start />2<milestone-end /> new supplementary figure panels.

Major points(1) The authors isolate disseminated tumor cells (DTC) and metastasis from tissues where these cells co-exist. It is assumed but not proven that the solitary DTCs came there from the pancreas and remained solitary while others grew out to form the profiled metastasis. However, the profiled DTCs could have separated from the neighboring metastasis large or small. PanC cells can be quite motile as shown by the senior author before and can move long distances (Nature. 2018 Sep;561(7722):201-205.). Further, a caveat for the interpretation of the profiles is that the presence of large metastasis may already alter the micro-environment explaining why the Seurat analysis cannot separate solitary and met populations – it is highly unusual that DTCs and metastasis do not differ in some PCA analysis. Without an exclusive solitary DTC state analysis the true nature of the DTCs may not be accurately resolved.To solve the above issue, the authors should perform two controls: (1) take advantage from the label retaining utility of Dendra2 or H2B-Dendra 2 (some papers are cited where this advantage was utilized) and photoconvert the cells in primary lesion. Then they could track and isolate red and green DTCs and later only green clusters and profile those using the same methods. (2) Option 1 is the best as it uses fate mapping reporting on quiescence and proliferation. But an alternative option is that the authors excise livers earlier and photoconvert solitary DTCs when there are still no metastasis. This would be very informative to distinguish the profiles of solitary DTCs in the context of low burden and more homeostatic liver vs. a high burden disease liver. If there are no differences between the DTCs then the isolation of stage IV-like DTCs becomes useful to profile those even in patients. But if there are differences then it may reveal how an earlier stage liver differs in the DTC populations from a late progression liver, information that would be very useful. Such analysis would enhance the paper as it remains limited in mechanism identification and strength of the data and phenotypes described.

We agree that our data cannot unambiguously assign the source of the metastatic cells as immediate descendants of the primary tumor as opposed to already-established metastases. To avoid confusion, and apply more precise terminology, we now refer to the small metastatic lesions as “microcolonies” rather than DTCs.

The absence of clear separation in the Seurat analysis also surprised us. However, this result is consistent with findings from a previous study performed by our lab (Aiello et al., 2016). In that study, we characterized phenotypic and molecular changes associated with metastatic growth (small vs. large metastases) and found that the fraction of cycling cells is similar across the spectrum of metastatic disease. Furthermore, these results are consistent with a recent study comparing primary breast cancer cells with matched micrometastatic nodules (Davis et al., 2020). That study reported little to no separation by PCA analysis but did identify differences in specific pathways. These results suggest that transcriptional differences that distinguish cells at different stages of metastatic progression may be insufficiently diverse to drive global differences that would be detectable on a PCA plot. In the revised manuscript, we have clarified this limitation of the study namely that our comparison reflects differences between small and large metastatic lesions rather than differences between newly arrived DTCs and advanced metastases.

Finally, we thank the reviewer for suggesting several experiments that could potentially help resolve these issues. We note that “option 1” has previously been used to great advantage to dissect events in early dissemination of breast cancer (Fluegen et al., 2017). However, either approach would require substantial work (>1 year) comparable in scope to that performed for the initial study. As an alternative, we turned to the autochthonous KPCY model to test whether our observation that microcolonies exhibit a pro-inflammatory profile holds true in the setting of stochastically arising micro-metastases. IHC analysis of KPC mice that have no overt metastatic disease revealed that nuclear NFkB and phospho-c-Jun staining is indeed prominent in the micro-metastases of these mice. These new data are included in the revised manuscript (Figure 4—figure supplement 1B,C).

(2) About the proliferative phenotype attributed to DTCs. Although single cell analysis suggests a non-proliferative phenotype for DTCs, no difference is observed in Ki67 staining. Then the authors show that isolated cells proliferate at different rates depending on their origin: DTCs were slower in increasing the number of cells in their colonies compared to cells isolated from metastatic colonies. On this single experiment, DTCs exhibit reduced propensity to proliferate, which for the authors was sufficient to suggest that these cells may be primed for cell cycle arrest. This conclusion needs more supporting information since no other marker of cell cycle arrest is observed. The authors see an increase cell cycle signatures in macrometastasis, but no evidence of cell cycle arrest, whether in gene signatures or cell culture experiments, is shown. What about phospho-histone 3 or P-Rb (by IF) or p27, p21, p16, p57 (IF or qPCR)? The re-culturing, while a nice assay, really shows no striking differences. There is a short lag difference that could be explained by adaptation or resolving DNA damage or alterations in the actin cytoskeleton as hinted from the ceacam and stard13 upregulation. The genes shown in Figure 3 are interesting but not classical negative regulators of growth.

We agree that more evidence is needed to define the native proliferation propensity of the cells in vivo and thank the reviewers for their suggestions. To further shed light on the propensity for cell cycle arrest, we conducted comprehensive profiling of the proposed panel of cell cycle markers by immunofluorescence, which is now included in the revised manuscript (Figure 3D-H). In agreement with reduced proliferation ex vivo, tumor cells comprising microcolonies exhibited reduced staining for phospho-histone H3 (Figure 3D), indicative of reduced mitosis. Likewise, staining intensity of phospho-RB+ cells is consistently lower in microcolonies (Figure 3E,G) while cells expressing p21 are more abundant in microcolonies (Figure 3H). (Results from our staining of P16, P27, and P57 indicated lack of nuclear expression in any of the tumor cells.) Taken together, these new data support the conclusion that tumor cells in microcolonies have reduced cell cycle activity compared to macrometastases in situ.

(3) Application of DTC and macrometastasis scores to TCGA data and survival. This is an unusual comparison. The primary tumor is already a growing proliferative mass with a strong stromal component in a different microenvironment. The comparison reveals probably only proliferation in the pancreas or in overt metastasis, which appears to be true as the curves only separate in the first 20 months arguing that the authors are looking at effects on growth regardless of the DTC or macro-metastasis state. Perhaps metastasis free proportion over time should be tested but more importantly the signatures from the requested experiments should be tested.

The rational for comparing the signatures of microcolonies (DTCs) and macrometastases to survival of PDAC patients was to explore how enrichment for signatures that characterize microcolonies (inflammatory) and macrometastases (biosynthetic/proliferative) would perform when applied to primary tumors. The results suggest that enrichment for the inflammatory signature in a primary tumor is at least as hazardous as having a high biosynthetic one. We agree, however, that it is impossible to draw conclusions about the behavior of differently sized metastatic lesions based on this analysis, and consequently we have removed these data from the revised manuscript.

Is there RNAseq from metastasis available to explore these mechanisms? One would assume that the profiles of the DTCs may not appear there but the macro-metastasis would?

As suggested, we identified one dataset (Lin et al., 2020) comprising matched primary tumors and macrometastases. (As expected, no datasets for microcolonies could be identified.) We then asked whether the genes showing higher expression in microcolonies in our datasets exhibited higher or lower expression in macrometastases as compared to the matched primary tumors in Lin et al. As can be seen from the UMAP plots in Author response image 1, genes that our study found to be enriched in microcolonies showed overall lower expression in macrometastases comparing to primary tumors. Specifically, some of the NF-κB transcription we found to be highly enriched in microcolonies (e.g. CXCL2, PTGS2, and INHBA) were largely absent from macrometastases in the human dataset. In contrast, genes that were enriched in macrometastasis in our dataset exhibited comparable or greater expression in human macrometastases as compared to matched primary tumors. These results overall support our observation that the low pro-inflammatory phenotype observed in our microcolonies is uncommon in macrometastasis. Because these data do not directly compare microcolonies with macrometastases, we have not included them in the revised manuscript but merely present them here for the reviewer. Finally, we note that a study comparing mouse pancreatic tumor CTCs with primary tumors derived from KPC mice (Ting et al., 2014) found that TNFa-mediated NF-κB signaling was the most highly enriched gene category in CTCs, providing independent evidence that this pathway may be active at the earliest stages of metastatic colonization. This similarity between our findings and those of Ting et al. has been noted in the revised manuscript.

Regarding the survival analyses: Were these controlled for other patient characteristics? How do the authors explain that enrichment of DTC-associated genes predicts poor survival in patients as opposed to those genes enriched in macromets? One may think the alternative is more likely. This seems to agree with the finding that DTCs seem to have an hypoproliferative phenotype compared to macromets – so how do they influence survival?

As noted above, we agree that it is hard to reach strong conclusions by applying gene signatures from differently sized metastatic lesions to a primary tumor dataset. Consequently, we have removed the survival analysis from the manuscript.

(4) Figure 4: This experiment is negatively affected by the same issues as in Point 1. The DTCs could be derived from the metastasis and not really true originally arrived DTCs. This experiment would be more impactful showing that H2B-dendra photoconverted cells from the primary site are eliminated by Abs to the chemokines or knock-down of its receptors. This would be useful and one could imagine how patients with PCa may receive prophylactic treatments to reduce DTCs based on unique biology. In stage IV disease it is probably meaningless unless some drug causes massive reduction in overt metastatic disease.

As noted above (point 1), our revision makes it clear that our comparison is between small and large metastases (without distinguishing the source of the small metastases). We agree with the reviewer’s interpretation of the value of targeting DTCs in early-stage disease. In line with this, it was recently reported that preoperative inhibition of NF-κB reduces the risk of recurrence in a PDAC mouse model (Saito et al., 2019). This reference has been added to the revised manuscript.